# Set7 mediated Gli3 methylation plays a positive role in the activation of Sonic Hedgehog pathway in mammals

Lin Fu[1], Hailong Wu[1], Steven Y Cheng[2], Daming Gao[1], Lei Zhang[1,3], Yun Zhao[1,3]*

[1]State Key Laboratory of Cell Biology, CAS Center for Excellence in Molecular Cell Science, Innovation Center for Cell Signaling Network, Institute of Biochemistry and Cell Biology, Shanghai Institutes for Biological Sciences, Chinese Academy of Sciences, University of Chinese Academy of Sciences, Shanghai, China; [2]Department of Developmental Genetics, School of Basic Medical Sciences, Nanjing Medical University, Nanjing, China; [3]School of Life Science and Technology, ShanghaiTech University, Shanghai, China

**Abstract** Hedgehog signaling plays very important roles in development and cancers. Vertebrates have three transcriptional factors, Gli1, Gli2 and Gli3. Among them, Gli3 is a very special transcriptional factor which closely resembles *Cubitus interruptus* (Ci, in *Drosophila*) structurally and functionally as a 'double agent' for Shh target gene expression. Here we show that Gli3 full-length, but not the truncated form, can be methylated at K436 and K595. This methylation is specifically catalyzed by Set7, a lysine methyltransferase (KMT). Methylation at K436 and K595 respectively increases the stability and DNA binding ability of Gli3, resulting in an enhancement of Shh signaling activation. Furthermore, functional experiments indicate that the Gli3 methylation contributes to the tumor growth and metastasis in non-small cell lung cancer in vitro and in vivo. Therefore, we propose that Set7 mediated methylation is a novel PTM of Gli3, which positively regulates the transactivity of Gli3 and the activation of Shh signaling.

*For correspondence: yunzhao@sibcb.ac.cn

**Competing interests:** The authors declare that no competing interests exist.

## Introduction

Hedgehog (Hh) signaling plays critical roles in embryonic development and tumor growth (*Chen et al., 2007*; *Hooper and Scott, 2005*; *Ingham and McMahon, 2001*; *Nieuwenhuis and Hui, 2005*; *Yao and Chuang, 2015*). Its misregulation leads to many types of cancers (*Gialmanidis et al., 2009*; *Hui and Angers, 2011*; *Jiang and Hui, 2008*). Hh pathway is activated when Hh ligands (Shh, Ihh or Dhh) bind to their repressive receptor Patched (Ptc), a twelve-transmembrane protein, leading to alleviation of its repression on the signaling transducer, Smoothened (Smo), a seven-transmembrane protein. The unleashed Smo can then activate the Gli transcription factors, resulting in the expression of downstream target genes, like *Gli1* and *Ptch1*. In vertebrate, there are three Gli proteins, Gli1, Gli2 and Gli3. While Gli1 acts as a transcriptional activator and amplifies the Shh signal in a positive feedback loop, Gli2 and Gli3 mainly function as the Shh-regulated transcriptional activator and repressor. Among them, Gli3 draws considerable attentions due to its structural and functional similarity to Ci. Like Ci, Gli3 plays critical roles in modulating the switch-on and -off of Shh signaling. In the absence of Shh signaling, Gli3 is partially proteolyzed into a truncated transcriptional repressor (Gli3[R]) to block the expression of downstream target genes. In the presence of Shh signaling, Gli3 is in a full-length transactivation form and activates the expression of downstream genes like *Gli1* and *Ptch1* (*Dai et al., 1999*; *2002*).

**eLife digest** Cells in mammals need to be able to communicate with each other to enable them to work together in tissues and organs. A signaling pathway called Hedgehog signaling plays a crucial role in carrying information between cells in developing embryos, but if it is active at other times it can also promote the development of cancers.

The Hedgehog signaling pathway regulates the activity of several proteins, including one called Gli3. When the Hedgehog signaling pathway is not active, Gli3 is able to switch off certain genes in the cells. On the other hand, when the signaling pathway is active, Gli3 changes shape so that it is able to activate its target genes instead. It is thought that this shape change is triggered by the addition (or removal) of chemical tags to Gli3. So far, researchers have reported that several different types of chemical tags can modify the activity of Gli3. However, it is not clear whether another type of chemical tag – known as a methyl tag – is involved in regulating Gli3.

Fu et al. studied Hedgehog signaling in mice. The experiments show that an enzyme called Set7 can modify Gli3 by adding methyl tags to certain sites in the protein. This modification makes the protein's structure more stable and helps it to bind to the target genes. Further experiments show that these methyl groups contribute to the progression of lung cancer. Fu et al.'s findings expand our understanding of how chemical tags can alter the cells' response to Hedgehog signaling activity. Future challenges are to understand exactly how Set7 and Gli3 interact and to develop drugs that can block this interaction, which may have the potential to treat cancer.

There is substantial evidence demonstrating that post-translational modifications (PTMs) play important roles in regulating protein functions, especially in regulating the transactivity of transcriptional factors (*Filtz et al., 2014*; *Huang et al., 2013*; *Inuzuka et al., 2011*; *2012*; *Liu et al., 2014*; *Munro et al., 2010*). Like many other transcriptional factors, Gli3 is also subjected to various PTMs. Among them, phosphorylation and ubiquitination modifications are well characterized in regulating the transactivity of Gli3. For example, in the absence of Shh signal, Gli3 is phosphorylated by PKA, GSK3 and CKI, and subsequently ubiquitinated by $SCF^{Slimb/\beta-TrcP}$ for partial proteolyzation to confer it trans-repressive activity (*Chen et al., 2009*; *Hsia et al., 2015*; *Tempé et al., 2006*; *Wang et al., 2000*; *Wang and Li, 2006*; *Zhang et al., 2009*). Whether other PTMs are involved in the regulation of Gli3 transactivity remains elusive.

Protein methylation is one of the most common PTMs and plays an important role in regulating the transduction of signaling pathways, like MAPK, BMP, WNT, Hippo and JAK-STAT (*Bikkavilli and Malbon, 2012*; *Kim et al., 2013*; *Mazur et al., 2014*; *Oudhoff et al., 2013*; *Viña et al., 2013*). Protein methylation typically happens on arginine or lysine residues catalyzed by peptidylarginine methyltransferases (PRMTs) or lysine methyltransferases (KMTs) respectively. So far, near 50 KMTs and 9 PRMTs had been identified in human genome (*Biggar and Li, 2015*). Among them, Set7 is one of the most studied KMTs, regarding its pivotal role in methylation of non-histone proteins. Although Set7 was first identified as a histone lysine methyltransferase specifically for Histone 3 lysine 4 mono-methylation, an epigenetic marker associated with transcriptional activation (*Nishioka et al., 2002*; *Wang et al., 2001*), accumulating evidence indicates that methylation of non-histone proteins including P53, P65, TAF10 and so on is the major biological function of this enzyme (*Biggar and Li, 2015*; *Chuikov et al., 2004*; *Ea and Baltimore, 2009*; *Yang et al., 2009*). Set7 mediated methylation of Lys372 in P53 increases its stability, resulting in the induction of P53 target genes (*Chuikov et al., 2004*). P65 can be methylated by Set7 at Lys37 which enhances the DNA binding and improves the expression of NF-κb target genes (*Ea and Baltimore, 2009*). Previous sequence alignments of the methylated sites on the initial substrates of Set7 revealed a predicted consensus sequence motif for Set7: (K/R)-(S/T/A)-**K**-X (*Couture et al., 2006*). Besides, a recent peptide-array based analysis redefined this recognition motif to: (G/R/H/K/P/S/T)-(K>R)-(S>K/Y/A/R/T/P/N)-**K**-(Q/N)-(A/Q/G/M/S/P/T/Y/V) (*Dhayalan et al., 2011*), which dramatically expands the putative targets of Set7.

Here, we report that Gli3 full-length, but not the Gli3 repression form, can be methylated at the K436 and K595 sites in vivo and in vitro. This methylation is specifically catalyzed by Set7. Moreover,

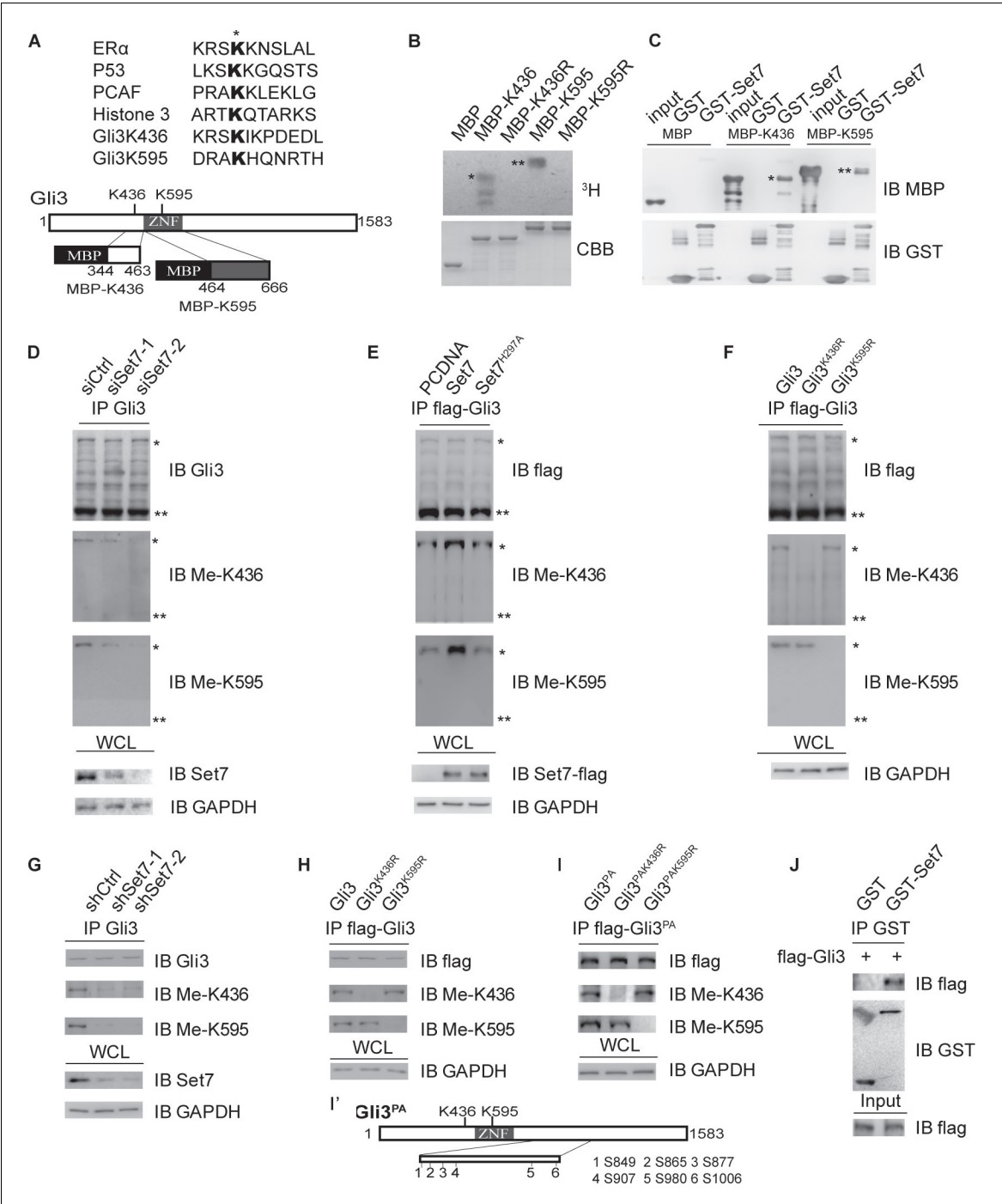

**Figure 1.** Set7 methylates Gli3 full-length in vivo and in vitro. (**A**) Sequence alignment of the reported Set7 substrates with Gli3K436 and Gli3K595 (upper). Schematic representation of Gli3 protein and the truncated peptide used in in vitro methylation assay (**B**) and GST pull-down assay (**C**) (lower). (**B**) In vitro methylation assay with $^3$H-S-adenosine-methionine ($^3$H-SAM), bacteria purified Set7 and MBP fusion protein. * and ** represent the MBP-K436 and MBP-K595 respectively. (**C**) GST pull-down assay using GST-Set7 and MBP tagged Gli3 truncated fragments described in (**A**). * and ** represent the MBP-K436 and MBP-K595 respectively. (**D–F**) Western blot of immunoprecipitates (top three panels) and lysates (bottom) from HEK293T cells expressing indicated siRNAs or proteins. * and ** represent the full-length and repressor forms of Gli3 respectively. (**G–I**) Western blot of immunoprecipitates (top three panels) and lysates (bottom) from NIH-3T3 cells stably expressing indicated shRNAs or proteins. (**I'**) Schematic representation of 6 PKA targeted serines which were mutated to nonphosphorylatable alanines in Gli3$^{PA}$. (**J**) GST pull-down assay using GST-Set7 and flag-Gli3 in NIH-3T3 cells in the presence of Shh. Ctrl, Control. Me-K436, antibody anti methylated Gli3-K436; Me-K595, antibody anti methylated Gli3-K595. WCL, whole cell lysis. The protein level of Gli3 in (**D–I**) are normalized to the same.

*Figure 1 continued on next page*

*Figure 1 continued*

The following figure supplements are available for figure 1:

**Figure supplement 1.** The mass spectrometry results show the methylation modification in the full-length Gli3 on K436 and K595.
**Figure supplement 2.** Sequence alignment of K436 and K595 sites of Gli3 in different species Sequence alignment of the methylation sites K436.
**Figure supplement 3.** The methylation antibodies can specifically recognize the mono methylated Gli3 peptides.
**Figure supplement 4.** The methylation antibodies can specifically recognize the mono methylated Gli3 full length in embroynic lung Indicated tissues from mouse embryos (14.5 dpc) were isolated and lysed.

the methylation modifications on K436 and K595 respectively increases the stability and the DNA binding capacity of Gli3, resulting in enhanced activation of Shh signaling pathway. Furthermore, we demonstrate that this Set7 mediated Gli3 methylations contribute to the tumor growth and metastasis in non-small cell lung cancer in vitro and in vivo. These findings expanded our understanding of PTM-directed Gli3 transactivity regulation, and implied a therapeutic potential of Set7 in treating tumors dependent on Shh signaling.

## Results

### Set7 methylates Gli3 full-length but not the repression form at K436 and K595 sites in vitro

Given that the transcriptional activity of Gli3 is orchestratedly regulated by multiple PTMs, such as phosphorylation and ubiquitination, and that protein methylation plays an important role in regulating several key signaling pathways, we sought to examine whether Gli3 can be post-translationally modified by methylation. We performed a mass spectrometry analysis of flag-tagged Gli3 from the cell lysate of HEK293T. This mass spectrometry analysis showed two methylation modifications on Gli3 K436 and K595 residues (*Figure 1—figure supplement 1*). By comparing the flanking sequence of K436 and K595 with reported Set7 substrates, such as ERα (*Subramanian et al., 2008*), P53 (*Chuikov et al., 2004*), PCAF (*Masatsugu and Yamamoto, 2009*) and Histone 3 (*Wang et al., 2001*), we found strong similarities among them (*Figure 1A*, upper panel), suggesting the possible involvement of Set7 in methylation of these two residues. Interestingly, these methylation signals were exclusively present in the Gli3 full-length but not the truncated repression form according to the mass spectrometry result (*Figure 1—figure supplement 1*). Through sequence alignments, we found that these two sites in Gli3 are evolutionarily conserved in many species (*Figure 1—figure supplement 2*). To further test if the methylations on K436 and K595 are catalyzed by Set7, in vitro methylation assays were performed by incubating MBP-tagged peptides of Gli3, MBP-K436 (amino acids from 344 to 463 of Gli3) or MBP-K595 (amino acids from 464 to 666 of Gli3) with GST-Set7 and [3]H-SAM (*Figure 1A*, lower panel). As shown in *Figure 1B*, Set7 can specifically methylate the peptides, MBP-K436 and MBP-K595, whereas the substitution of K436 or K595 with arginine (K436R or K595R) abolished Set7 mediated Gli3 methylation. In addition, GST pull-down assays further indicated the interaction between Set7 and Gli3 (*Figure 1C*).

### Set7 methylates Gli3 at K436 and K595 in vivo

To facilitate studies on the Set7-mediated methylation of Gli3 in vivo, we generated polyclonal antibodies that specifically recognize monomethylated Gli3-K436 or Gli3-K595, respectively (*Figure 1—figure supplement 3A*). Dot plot assays confirmed the specificity of those antibodies (*Figure 1—figure supplement 3*). Consistent with the results of mass spectrometry, immunoprecipitation (IP) of endogenous Gli3 in HEK293T cells followed by immunoblot showed that only the full-length Gli3 contains the methylation signals on Gli3-K436 and Gli3-K595 (*Figure 1D*). The methylation signals were decreased upon Set7 knockdown, indicating that methylation on these two sites of Gli3 is mediated by Set7 (*Figure 1D*). Moreover, overexpression of Set7 drastically improved the methylation levels whereas the catalytically inactive mutant of Set7, Set7[H297A] (*Wang et al., 2001*), had no

effect on those methylations of Gli3 (*Figure 1E*). Furthermore, Set7 failed to methylate Gli3$^{K436R}$ or Gli3$^{K595R}$, the arginine substitution mutants of Gli3, in vivo (*Figure 1F*).

Since the methylation happens exclusively in the full-length Gli3, which acts as a principal transcription activator of Shh pathway in response to Shh signal (*Dai et al., 1999*), we then adopted, NIH-3T3, a cell line fully responsive to the Shh signaling, to perform our further investigation. Cells with stable expression of individual unrelated Set7 shRNAs showed significantly reduced methylation signals on K436 and K595 of Gli3 compared to cells expressing luciferase control shRNA (*Figure 1G*). In addition, the Gli3 arginine substitution mutants, Gli3$^{K436R}$ and Gli3$^{K595R}$, showed abolished methylation signals when blotted with the matched antibodies (*Figure 1H*). Given that only the full-length Gli3 can be methylated, we began to use a Gli3$^{PA}$ mutant, in which the six phosphorylatable serines in the PKA target clusters were mutated to nonphosphorylatable alanines, resulting in the resistance of Gli3$^{PA}$ to be processed into Gli3$^{R}$ (*Niewiadomski et al., 2014*) (*Figure 1I'*). Similar to the result in *Figure 1H*, the methylation signals can be detected in Gli3$^{PA}$ but not in its arginine substitution mutants, Gli3$^{PAK436R}$ and Gli3$^{PAK595R}$ when blotted with respective antibodies (*Figure 1I*). Given the Shh signaling plays an important role in the development of multiple embryonic tissues of mice (*Ingham and McMahon, 2001*), we started to examine whether these methylation signals exist in the Gli3 in several responsive mouse embryonic tissues, such as brain, lung, skin, skeleton and gut. Indicated tissues were collected from embryos 14.5 days post coitum (dpc) and Gli3 methylation signals were detected by western blot using anti-Gli3 antibody, Me-K436 antibody and Me-K595 antibody, respectively. As shown in *Figure 1—figure supplement 4*, the methylation signals on Gli3 full length were only detected in embryonic lung tissues. Since Gli3 full-length can be methylated by Set7 in vivo and in vitro, we then examined whether Set7 interacts with Gli3. GST pull-down assays indicated that GST-tagged Set7 can successfully pull down the flag-tagged Gli3 full-length of NIH-3T3 cells in the presence of Shh (*Figure 1J*). Therefore, we demonstrate that Set7 can methylate the full-length, but not the repression form of Gli3, at K436 and K595 residues, indicating that Gli3 is a novel substrate of Set7.

## Set7 positively regulates the activation of Shh signaling by methylating Gli3

Set7 can modulate the activation of multiple signaling pathways, such as NF-κb, JAK-STAT, we sought to examine whether Set7 also regulates Shh signaling. We performed Shh luciferase reporter assays (*Kinzler and Vogelstein, 1990*) in the condition of Set7 knockdown. In the absence of Shh, downregulation of Set7 had no effect on the luciferase activity. In contrast, Set7 knockdown dramatically decreased the luciferase activity in the presence of Shh, suggesting that Set7 plays a positive role on Shh pathway activation but has no effect on the basal level of Shh signaling (*Figure 2A and A'*). In line with this finding, overexpression of Set7 but not its inactive mutant Set7$^{H297A}$ can significantly enhance the luciferase activity in the presence of Shh (*Figure 2B*), suggesting that its methyltransferase activity is required for Set7 mediated Shh activation. Furthermore, Set7 mediated increase of luciferase activity was in a dose-dependent manner (*Figure 2C*). We then examined whether Set7 knockdown affects the expression of endogenous Shh target genes, like *Gli1* and *Ptch1*. In the presence of Shh, NIH-3T3 cells expressing Set7 siRNAs showed significant reduction in Set7 protein levels and a parallel decrease in *Gli1* and *Ptch1* mRNA levels compared to cells expressing control siRNAs (*Figure 2D–F*). In addition, Gli1 protein level was also decrease in cells with Set7 knockdown (*Figure 2F*). It's of interest that either Shh or SAG, the agonist of Shh pathway, directed activation of Shh signaling can upregulate Set7 levels, whereas Gli1 knockdown can lead to a decrease of *Set7* mRNA levels, suggesting the existence of a potential positive feedback loop between Set7 and Shh signaling (*Figure 2A'* and *Figure 2—figure supplement 1*). In addition, NIH-3T3 cells treated with Shh showed greatly increased methylation signals on Gli3 which may be probably due to increased Gli3 full-length or/and Set7 levels (*Figure 2A'* and *Figure 2—figure supplement 2*), suggesting that the methylation signals of Gli3 is regulated by Shh input. Taken together, these findings indicate that Set7 can positively regulate the activation of Shh signaling.

Next, we started to investigate whether the positive regulation of Set7 on Shh signaling relies on its methylation on Gli3. As shown in *Figure 2G*, although Set7 knockdown can suppress the activation of Shh signaling, this inhibition was abolished when the Gli3 level was depleted by RNAi. In contrast, depletion of both Gli1 and Gli2 in MEF cells showed no effect on releasing such inhibition due to Set7 knockdown (*Figure 2—figure supplement 3*). These findings apparently indicate the

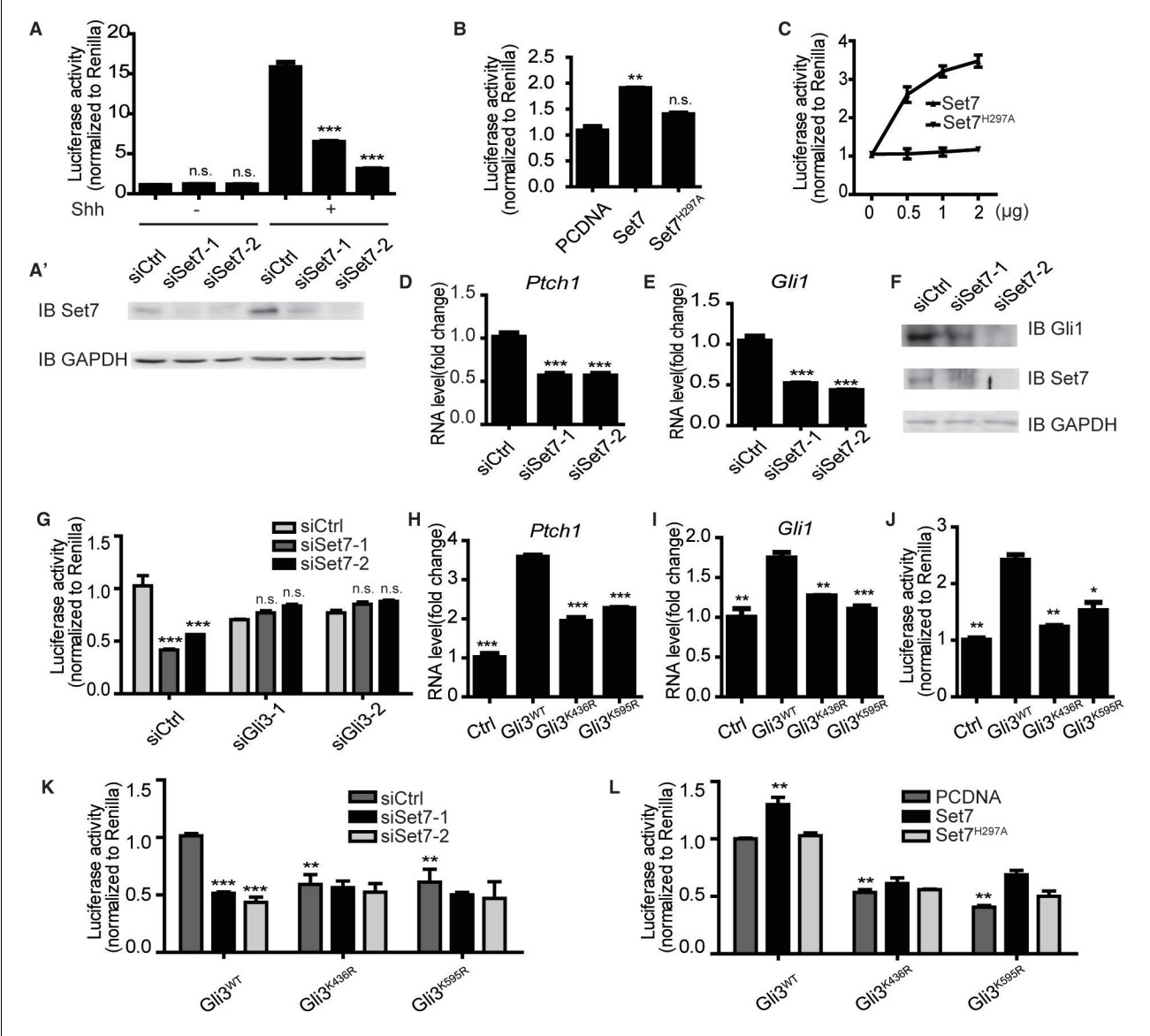

**Figure 2.** Methylation by Set7 promotes the Sonic Hedgehog signaling activity. (**A–C**) Shh luciferase reporter assays in NIH-3T3 cells expressing indicated siRNAs or proteins. (**A′**) Set7 level changes in NIH-3T3 cells expressing indicated Set7 siRNAs. n.s. $p>0.05$, ***$p<0.005$ versus siCtrl respectively in (**A**). (**D** and **E**) qPCR to detect Shh target gene, *Gli1* and *Ptch1*, in Shh treated NIH-3T3 cells after Set7 knockdown. ***$p<0.005$ versus siCtrl. (**F**) Western blot to detect protein level changes of Set7 and Gli1 in (**D** and **E**). (**G**) Shh luciferase reporter assay in Shh treated NIH-3T3 cells transfected with indicated siRNAs. ***$p<0.001$, n.s. $p>0.05$ versus siCtrl siCtrl respectively. (**H–J**) mRNA level changes of *Gli1* and *Ptch1* (**H** and **I**) and Shh luciferase activity changes (**J**) in Shh treated NIH-3T3 cells transfected with unmethylatable mutants, Gli3$^{K436R}$ and Gli3$^{K595R}$. *$p<0.05$, **$p<0.01$, ***$p<0.001$ versus Gli3WT (**K** and **L**) Shh luciferase assay in Shh treated NIH-3T3 cells expressing indicated siRNAs or/and proteins., **$p<0.01$, ***$p<0.001$ versus siCtrl Gli3WT or PCDNA Gli3WT respectively.. Results in (**A–E**, **G–L**) are shown as mean ± SEM (n=3). All the qPCR results are normalized to GAPDH. Ctrl, Control.

The following figure supplements are available for figure 2:

**Figure supplement 1.** Sonic hedgehog signaling increases the mRNA level of Set7.

**Figure supplement 2.** The methtylation of Gli3 is regulated by the Shh signaling Western blot of endogenous Gli3 immunoprecipitates from NIH-3T3 cells with or without Shh treatment. * and ** represent the full-length and repressor forms of Gli3 respectively.

**Figure supplement 3.** In the Gli1$^{-/-}$; Gli2$^{-/-}$ MEF cells, Set7 still regulates the Shh signaling.

*Figure 2 continued on next page*

Figure 2 continued

**Figure supplement 4.** Non-methylation mutants show reduced Gli1 and Ptch1 mRNA level, and decreased Shh luciferase activity.

requirement of Gli3 on Set7 mediated Shh activation. Furthermore, compared to Gli3$^{WT}$, the mRNA levels of *Gli1* and *Ptch1* and the Shh luciferase activity were decreased in cells expressing Gli3$^{K436R}$ and Gli3$^{K595R}$, suggesting that Set7-directed Gli3 methylation plays an important role in maintaining Shh activation (*Figure 2H–J*). In consistent with this finding, cells expressing the Gli3$^{PA}$ point mutants, Gli3$^{PAK436R}$ and Gli3$^{PAK595R}$, showed decreased Shh luciferase activity and levels of *Gli1* and *Ptch1* transcripts compared to those with Gli3$^{PAWT}$ expression (*Figure 2—figure supplement 4*). To further confirm this notion, Shh luciferase reporter assays were performed. Either Set7 down-regulation or upregulation can lead to the reduction or the induction of Shh luciferase activity respectively in cells with Gli3$^{WT}$ expression but not in cells expressing Gli3$^{K436R}$ and Gli3$^{K595R}$ (*Figure 2K and L*). Thus, our findings strongly suggested that Gli3 methylation at K436 and K595 sites are required for Set7-mediated enhancement of Shh signaling activation.

## Methylation by Set7 improves the stability and DNA binding ability of Gli3 full-length

Gli3 can be regulated at multiple levels to affect the transduction of Shh signaling, such as trafficking to the primary cilium, changes of protein stability or binding ability to the promoters of target genes. In mammal, the primary cilium (cilia) is an essential organelle for Shh signaling transduction. Gli3 traffics in and out of the cilia all the time (*Figure 3—figure supplement 1A and B*). In the presence of Shh, Gli3 initially accumulates in the cilium tip (*Figure 3—figure supplement 1B*) and then functions as Gli3$^A$ to bind to the promoter of *Gli1* to further transduce the signaling (*Chen et al., 2009*; *Dai et al., 1999*; *2002*; *Wen et al., 2010*; *Zhang et al., 2009*). In order to verify whether Set7 mediated Gli3 methylation affects its cilial translocation, we performed cilial staining in Shh treated NIH-3T3 cells with stable expression of Set7 shRNA, Set7 or Set7$^{H297A}$. As shown in *Figure 3—figure supplement 1C and D*, the Gli3 accumulation on the tips of cilia was comparable among those three groups, suggesting that the Set7-related changes of Shh activation are not due to the cilial translocation of Gli3.

It has been reported that Set7 can influence the stability of its methylated substrates like P53 (*Chuikov et al., 2004*). Given the importance of Gli3 stability on Shh signaling activation (*Tempé et al., 2006*; *Zhang et al., 2009*), we then tested whether Set7 can also affect the stability of Gli3. Western blot showed that knockdown of Set7 caused a significant reduction of endogenous Gli3 at protein levels (*Figure 3A*, left panel), whereas overexpression of Set7 improved the Gli3 protein level (*Figure 3A*, right panel). This Set7 mediated induction of Gli3 seems due to PTMs, because the mRNA level of Gli3 had no change regardless whether Set7 was knocked down or overexpressed (*Figure 3B*). To further confirm this point, we then tested whether Set7 can affect the protein levels of the unmethylatable Gli3 mutants, Gli3$^{K436R}$ and Gli3$^{K595R}$. As expected, exogenous Gli3$^{WT}$ had the same change as the endogenous Gli3. Interestingly, like the Gli3$^{WT}$, the protein levels of Gli3$^{K595R}$ were still subjected to the modulation of Set7, whereas the protein levels of Gli3$^{K436R}$ were not responding to either knockdown or overexpression of Set7 (*Figure 3C and D*). These findings suggested that the methylation at K436 but not K595 causes stability change of Gli3 (*Figure 3C and D*). Finally, the reduced half-life of Gli3$^{K436R}$, but not Gli3$^{WT}$ or Gli3$^{K595R}$ was defined by western blot in cells treated with cycloheximide (CHX) (*Figure 3E and F*). Therefore, these data indicated that Set7 mediated methylation on K436 increases the protein stability of Gli3.

Gli3 full-length is a transcription activator and can bind to the promoter of *Gli1* to further enhance the activation of Shh signaling (*Hui and Angers, 2011*; *Jiang and Hui, 2008*). To examine whether Set7 also affects the DNA binding ability of Gli3, chromatin immunoprecipitation (ChIP) assays were performed in NIH-3T3 cells in the presence of Shh. As shown in *Figure 3—figure supplement 2*, knockdown of Set7 greatly decreased the binding of Gli3 on *Gli1*'s promoter regions (*Figure 3G* and *Figure 3—figure supplement 2*). Although Shh treatment can almost completely block the processing of Gli3 full-length to Gli3$^R$, trace Gli3$^R$ background may still disturb the result of ChIP assays. In order to address this concern, we established NIH-3T3 cell lines with stable expression of C-terminal flag-tagged Gli3 (Gli3-flag). Thus, only the Gli3 full-length can be recognized by anti-flag

antibody. ChIP assays with anti-flag antibody demonstrated that Set7 knockdown markedly affected the binding of Gli3 on *Gli1*'s promoter regions (*Figure 3G–I*), suggesting that Set7 augments the DNA binding of Gli3 full-length. Given that K595 is on the zinc finger which is important for DNA binding of Gli3 (*Pavletich and Pabo, 1993*), it's conceivable that the Set7 mediated methylation on K595 may contribute to increased DNA binding of Gli3. Stable cell lines expressing Gli3$^{WT}$-flag, Gli3 mutants (Gli3$^{K436R}$-flag and Gli3$^{K595R}$-flag) were established. Comparable expression of flag-tagged Gli3 was detected in those three cell lines (*Figure 3J*). ChIP assays with anti-flag antibody showed dramatically impaired the DNA binding ability in Gli3$^{K595R}$-flag, whereas the Gli3$^{K436R}$-flag presented the similar DNA binding ability to Gli3$^{WT}$-flag (*Figure 3K*), suggesting that Set7 mediated methylation at K595 enhances the DNA binding ability of Gli3. In addition, the binding ability of Gli3 was also examined in the promoter regions of *Ptch1* and *Hhip* by ChIP assays. As shown in *Figure 3—figure supplement 3*, the binding of Gli3 to these regions was markedly impaired after Set7 knockdown. In conclusion, Set7 improved the protein stability and DNA binding ability of Gli3 through methylation of K436 and K595 sites respectively.

## The methylation of Gli3 positively regulates tumorigenesis in NSCLCs

Misregulation of Shh pathway has been reported in many types of cancers, including basal cell carcinoma, medulloblastoma and lung cancer (*Gialmanidis et al., 2009*; *Hui and Angers, 2011*; *Jiang and Hui, 2008*; *Yuan et al., 2007*). Since Set7 can augment Shh activation by increasing Gli3 stability and its DNA binding ability on the promoter regions of *Gli1*, we then determined whether this Set7-Gli3-Gli1 axis plays a role in tumor development. By comparison on the mRNA levels of *Set7* and *Gli1* between Beas2B, a normal bronchial epithelial lung cell line, and A549, a non-small cell lung cancer cell line, we found that both *Set7* and *Gli1* were extensively expressed in A549 cells (*Figure 4—figure supplement 1A*). Furthermore, the GEO database search (GSE10245) showed a positive association between *Set7* and *Gli1* in NSCLC tumor samples, suggesting the involvement of this Set7-Gli3-Gli1 axis in the development of NSCLCs (*Figure 4—figure supplement 1B*). To examine the existence of this regulation axis in A549 cells, we performed Shh luciferase reporter assays and qPCR to evaluate the activity of Shh signaling in A549 cells with Set7 knockdown. Depletion of Set7 displayed a decreased luciferase activity and reduced expression of *Gli1* and *Ptch1* (*Figure 4A and B*). Our above findings indicate that unmethylatable mutations at either K436 or K595 cause a significant reduction of Shh signaling activation (*Figure 2H–J*). To examine whether unmethylatable mutations at both sites will lead to a synergistic suppression on Shh signaling activation, we constructed a double arginine substitution mutant of Gli3, Gli3$^{KRKR}$, in which both K436 and K595 were mutated to arginine. To further examine whether Gli3 methylations at K436 and K595 are important for Set7-mediated Shh activation in A549 cells, we performed luciferase assays and qPCR. Compared to A549 cells with stable Gli3$^{WT}$ expression, reduced luciferase activity and transcripts of *Gli1* and *Ptch1* were detected in cells stably expressing Gli3$^{K436R}$, Gli3$^{K595R}$ or Gli3$^{KRKR}$ (*Figure 4C–E*). In addition, more severe suppression on luciferase activities and *Ptch1* mRNA, but not *Gli1* mRNA, was observed in cells expressing Gli3$^{KRKR}$, suggesting that the methylation at these two sites plays a synergistic role in regulating Shh signaling activation (*Figure 4D and E*). In line with the reduced Shh activation, MTT assays showed attenuated growth in cells stably expressing Gli3$^{K436R}$, Gli3$^{K595R}$ or Gli3$^{KRKR}$ compared to cells with Gli3$^{WT}$ expression (*Figure 4F*). Since anchorage-independent growth is tightly associated with tumor development, we then determined whether Set7-mediated Gli3 methylation affects this feature. The ability to form colonies in soft agar was dramatically impaired in stable transfectants with the expression of Gli3$^{K436R}$, Gli3$^{K595R}$ or Gli3$^{KRKR}$ compared to Gli3$^{WT}$ (*Figure 4G and G'*). In agreement with these in vitro growth assays, A549 cells with stable expression of Gli3$^{K436R}$, Gli3$^{K595R}$ or Gli3$^{KRKR}$ showed retarded tumor growth in a xenograft tumor model (*Figure 4H–J*).

A recent report indicates that the activation of Shh pathway is required for epithelial to mesenchymal transition (EMT) in NSCLCs (*Della Corte et al., 2015*). Combined with our results above, we started to examine whether this Set7-Gli3-Gli1 axis also plays a role in cell migration or invasion. To evaluate cell migration changes, transwell migration and wound healing assays were performed in A549 cells. As shown in *Figure 4K and L*, A549 cells with the expression of Gli3$^{K436R}$, Gli3$^{K595R}$ or Gli3$^{KRKR}$ had limited migration ability compared to Gli3$^{WT}$. Furthermore, matrigel chamber assays were performed and the invasion ability of A549 cells stably expressing Gli3$^{K436R}$, Gli3$^{K595R}$ or Gli3$^{KRKR}$ was substantially reduced (*Figure 4M*). Moreover, all the in vitro functional experiments in *Figure 4* were repeated in A549 cells stably transfected with Gli3$^{PAWT}$, Gli3$^{PAK436R}$, Gli3$^{PAK595R}$ or

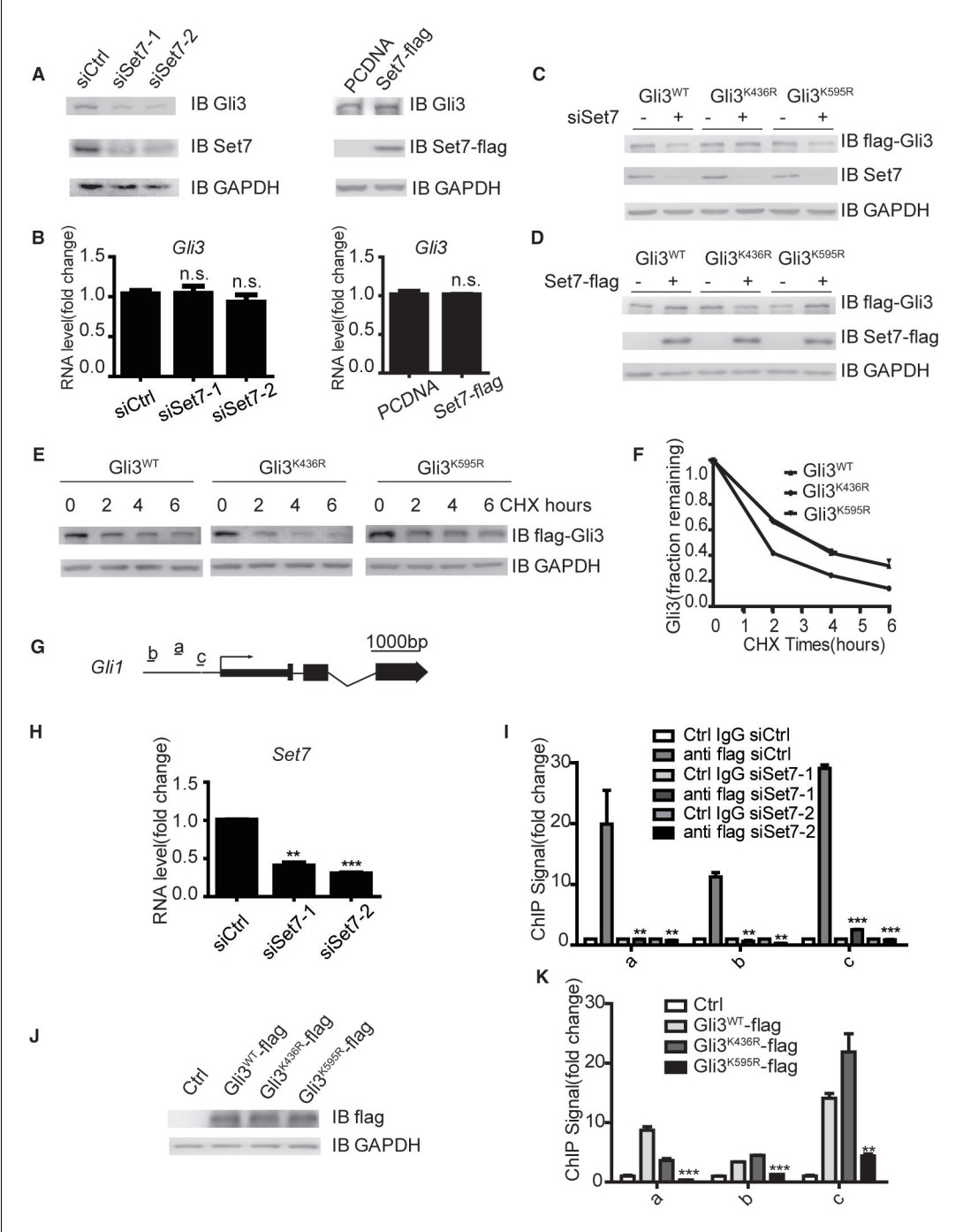

**Figure 3.** Set7 improved the stability and DNA binding ability respectably through K436 and K595 methylation. (**A** and **B**) Western blot (**A**) and qPCR (**B**) of Gli3 in the condition of Set7 knockdown or overexpression in NIH-3T3 cells. n.s. p>0.05 versus siCtrl (left) or PCDNA (right) respectively. (**C** and **D**) Western blot of Gli3$^{WT}$, Gli3$^{K436R}$ or Glli3$^{K595R}$ in the condition of Set7 knockdown or overexpression in NIH-3T3 cells. (**E** and **F**) Western blot (**E**) and statistic (**F**) of Gli3$^{WT}$, Gli3$^{K436R}$ or Gli3$^{K595R}$ in NIH-3T3 cells treated with cycloheximide (CHX) for the indicated times. (**G**) Schematic representation of three Gli3 binding sites on the promoter regions of *Gli1*. (**H**) qPCR of Set7 in Gli3$^{WT}$-flag stable cell lines after Set7 knockdown in (**I**). **p<0.01, ***p<0.00 versus siCtrl. (**I**) ChIP-qPCR analysis using Ctrl IgG (grey) or anti-flag antibody (black) in NIH-3T3 cells transfected with Ctrl or Set7 siRNAs. ChIP signal levels are represented as fold change of input chromatin. (**J**) Western blot of flag tagged Gli3$^{WT}$ and mutants in the stable cell lines used in (**K**). (**K**) ChIP-qPCR analysis using anti-flag antibody in NIH-3T3 cells expressing Gli3$^{WT}$-flag, Gli3$^{K436R}$-flag or Gli3$^{K595R}$-flag. ChIP signal levels are represented as fold change of input chromatin. *p<0.05, **p<0.01, ***p<0.001, n.s. p>0.05 versus anti flag siCtrl (**I**) or Gli3$^{WT}$-flag (**K**). Data are represented as mean ± SEM (n=3). All the qPCR results are normalized to GAPDH. In all the experiments, cells are treated with Shh. Ctrl, Control.

*Figure 3 continued on next page*

*Figure 3 continued*

The following figure supplements are available for figure 3:

**Figure supplement 1.** Set7 does not influence the cilium translocation of Gli3.

**Figure supplement 2.** Set7 improves the DNA binding ability of Gli3 on Gli1 promoter.

**Figure supplement 3.** Set7 improves the DNA binding ability of Gli3 on the promoter regions of *Hhip* and *Ptch1* ChIP-qPCR assays at *Hhip*.

Gli3$^{PAKRKR}$, and showed similar results (*Figure 4—figure supplement 2*). It further confirms that the methylation on Gli3 full-length regulates tumorigenesis in NSCLCs.

## Discussion

In present study, we demonstrated that Set7 can exclusively methylate Gli3 full-length at K436 and K595 sites in vitro and in vivo. These methylations on Gli3 augment the activation of Shh signaling. To our knowledge, we for the first time identified Gli3 as a novel substrate of Set7 and demonstrated the involvement of Set7 in Shh activation. Furthermore, we showed that Set7 mediated methylation on K436 increases the protein stability of Gli3 whereas the methylation on K595 boosts the DNA binding of Gli3. Finally, functional experiments indicated that this Set7-mediated Gli3 methylation contributes to the tumor growth and metastasis in NSCLCs in vitro and in vivo.

Hh signaling is one of the most conserved signaling pathways. The silence or activation of Hh signaling is tightly regulated by various post-translational modifications (PTMs) at multiple levels. Gli3 is a key transcriptional factor determining the switch-off and -on of Shh signaling. The PTMs of Gli3 like phosphorylation (*Tempé et al., 2006*; *Wang et al., 2000*; *Wang and Li, 2006*) and ubiquitination (*Chen et al., 2009*; *Hsia et al., 2015*; *Zhang et al., 2009*) are largely involved in this process. For example, in the absence of Shh, Gli3 full-length is phosphorylated by PKA, GSK3 and CK1 and is subjected to SCF$^{Slimb/β-TrcP}$ mediated ubiquitination followed with partial proteasome degradation to generate a truncated repression form, Gli3$^{R}$ (*Tempé et al., 2006*; *Wang et al., 2000*; *Wang and Li, 2006*).

Protein methylation, as one of the most common PTMs, participates in the regulation of signaling transduction of multiple signaling pathways, like MAPK, BMP, WNT, Hippo and JAK-STAT (*Bikkavilli and Malbon, 2012*; *Kim et al., 2013*; *Mazur et al., 2014*; *Oudhoff et al., 2013*; *Xu et al., 2013*). Protein methylation is catalyzed by methyltransferase. Among them, Set7 draws tremendous attention to its predominant role in methylating non-histone proteins. In this study, we identify that Gli3 is a novel substrate of Set7. By methylating Gli3, Set7 positively regulates the activation of Shh signaling pathway.

It's of particular interest that, although Gli3$^{R}$ also has these two residues, K436 and K595, the Set7 mediated methylation solely occurs in Gli3 full-length. One possible explanation comes from the differential cellular localization of Gli3$^{R}$ and Gli3 full-length in the absence of Shh. A recent study demonstrates that the stability of Dnmt1 can be dynamically regulated by Set7 and demethylase, Lsd1 (*Wang et al., 2009*). Since Lsd1 is a nuclear protein, it is conceivable that Lsd1 can also remove the methylation modifications of Gli3$^{R}$, resulting in these signals undetectable either by mass spectrometry or western blot. In addition, we found increased Set7 levels in cells treated with Shh or SAG, suggesting a positive feedback loop between Set7 and Shh signaling. Thus, in the presence of Shh signaling, although Gli3 full-length also translocates into the nucleus subjected to Lsd1 mediated demethylation, increased Set7 could possibly compensate this demethylation effect, resulting in augment in Shh signaling. Therefore, it will be particularly interesting to examine whether Lsd1 and Set7 can dynamically regulate the activation of Shh signaling by demethylation and methylation of Gli3 respectively.

We demonstrated that Set7 methylates K436 and K595 of Gli3. The methylation on K436 increases the protein stability and the methylation on K595 enhances the DNA binding of Glli3. These two methylation modifications of Gli3 are not redundant, because arginine substitution in either point severely impairs the activation of Shh signaling, suggesting the role of Set7 in fine-tuning the transactivity of Gli3.

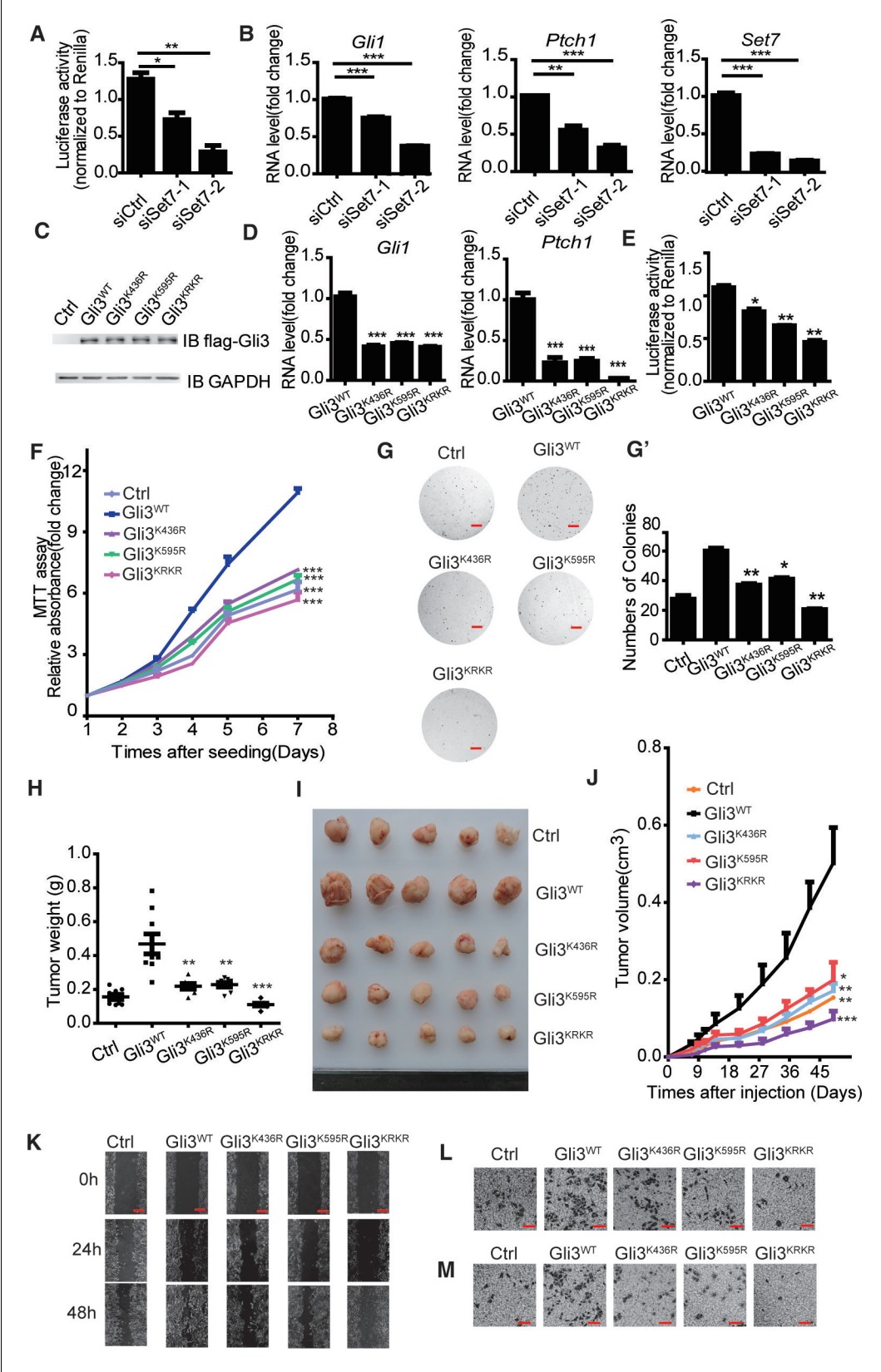

**Figure 4.** Methylation of Gli3 positively regulated the A549 proliferation and migration ability. (**A** and **B**) Shh luciferase reporter assay (**A**) and qPCR (**B**) of *Gli1, Ptch1* and *Set7* in A549 cells in the condition of Set7 knockdown. (**C**) Western of flagged Gli3^WT^, Gli3^K436R^, Gli3^K595R^ or Gli3^KRKR^ in A549 stable
*Figure 4 continued on next page*

*Figure 4 continued*

cells used in (D-M). (D and E) qPCR (D) of *Gli1* and *Ptch1* and Shh luciferase reporter assay (E) in A549 cells expressing Gli3$^{WT}$, Gli3$^{K436R}$, Gli3$^{K595R}$ or Gli3$^{KRKR}$. (F) MTT assays in A549 cells expressing indicated proteins. (G and G') Photography (G) and statistic results (G') of anchorage-independent growth assay in A549 cells expressing indicated proteins. (H and I) Statistic (H) and photography (I) results of tumor size in A549 cells expressing indicated proteins. (J) Tumor growth curves of A549 cells express indicated protein in a xenograft mouse model. (K–M) Migration ability evaluated by wound healing assay (K) and transwell migration assay (L). Invasiveness evaluated by matrigel invasion assay (M) in A549 cells expressing indicated proteins. All the qPCR results are normalized to GAPDH. Data in (D–H, J) were represented as mean ± SEM (n=3). *p<0.05, **p<0.01, ***p<0.001 versus Gli3$^{WT}$. Ctrl, Control.

The following figure supplements are available for figure 4:

**Figure supplement 1.** The involvement of this Set7-Gli3-Gli1 axis in the development of NSCLCs
**Figure supplement 2.** Methylation on the Gli3$^{PA}$ promoted the A549 tumorigenesis.

It has been recently reported that the activation of Shh signaling represses the canonical WNT signaling in colon epithelial cell differentiation, and that WNT pathway activation negatively regulates Shh pathway (*van den Brink et al., 2004*; *Yanai et al., 2008*), suggesting a mutual repelling crosstalk between these two pathways. Given that Set7 can inhibit WNT signaling by methylating β-catenin, and activate Shh signaling by methylating Gli3, it's very possible that Set7 is a key regulator involved in this WNT-Shh crosstalk.

Abnormal activation of Shh signaling contributes to tumor development by promoting cell proliferation and EMT in many cancer types. Our present study indicates that Set7 mediated Gli3 methylation is critical for this oncogenic role of Shh pathway because cancer cells expressing Gli3$^{K436R}$, Gli3$^{K595R}$ or Gli3$^{KRKR}$ showed severely reduced growth and metastasis in xenograft tumor models. This finding strongly suggests a therapeutic value of Ser7 in targeting Shh-dependent tumors, such as BCC, MB and Lung cancer.

## Materials and methods

### Plasmid construction

The Gli3 and Set7 expression plasmids were kindly provided by Dr. Chi-chung Hui and Dr. Min Wu respectively. The Gli3$^{PA}$ plasmid was obtained from Addgene (Plasmid# 226672). The unmethylatable mutants of Gli3, Gli3$^{K436R}$, Gli3$^{K595R}$, Gli3$^{KRKR}$, Gli3$^{PAK436R}$, Gli3$^{PAK595R}$, Gli3$^{PAKRKR}$, and the methylase mutant of Set7, Set7$^{H297A}$, were generated by site-directed mutagenesis kit (TOYOBO, Osaka, Japan). The DNA sequences of the mutants were verified by Sanger DNA sequencing. The GST-tagged expression plasmid, GST-Set7 and the MBP-tagged expression plasmids, MBP-K436, MBP-K595, MBP-K436R and MBP-K595R were cloned into PGEX3 vector (Promega, Madison, WI) and expressed in *E coil*.

### Cell culture, transfection and treatment

HEK293T, A549 and NIH-3T3 were purchased from Cell Bank, The Culture Collection, Chinese Academy of Sciences (CBTCCCAS). CBTCCCAS has provided certifications for the source, identity and mycoplasma contamination of these cell lines. HEK293T cells and A549 cells were cultured in DMEM medium (HyClone, Logan, UT) with 10% FBS (Gibco, Wellesley Hills, MA) and transfected with Lipofectamine2000 (Invitrogen, Waltham, MA) according to the manufacturer's protocol. NIH-3T3 cells were cultured in DMEM medium (HyClone, UT) with 10% NCS (Gibco) and transfected with Polyfect (QIAGEN, Hilden, Germany) according to the manufacturer's protocol. Recombinant Shh protein (R&D, Minneapolis, MN) or SAG (CALBIOCHEM, Darmstadt, Germany) was prepared in DMEM medium with 0.5% NCS and added to cells 24 hr after transfection. Gli1$^{-/-}$;Gli2$^{-/-}$ MEF cells (*Lipinski et al., 2008*) were kindly provided by Dr. Steven Y. Cheng.

### Immunoprecipitation and western blot

HEK293T cells were lysed using NP-40 buffer (50 mM Tris-Cl pH8.0, 0.1 M NaCl, 10 mM Sodium fluoride, 1 mM Sodium vanadate, 1% NP-40, 10% Glycerol, 1.5 mM EDTA, Protease Inhibitor

Cocktail). Cell lysates were rotated for 30 min at 4°C and cleared by centrifuge at full speed. Lysates were then incubated with M2 beads (Sigma-Aldrich, St Louis, MO) for 24 hr at 4°C. After 4-time wash (15 min/time) with NP-40 buffer, the protein samples were collected by boiling in 1×SDS loading buffer and subjected to standard SDS-PAGE and Western Blot. Primary antibodies used in this study: Mouse anti-flag (Sigma-Aldrich), Mouse anti-Set7 (Cell signaling, Danvers, MA), Goat anti-Gli3 (R&D), Rabbit anti-GAPDH (Sigma-Aldrich), Rabbit anti-monomethylated K436 (Me-K436, Abcolony, China), Rabbit anti-monomethylated K595 (Me-K595, Abcolony, China). Please also refer the figure legend of *Figure 1—figure supplement 3* for more detailed information. Secondary antibodies used in this study were purchased from Millipore Company.

## Mass spectrometry

Flag tagged Gli3 were transfected into the HEK293T cells. Cells were collected 48 hr after transfection. Then IP flag using M2 beads as described above. The protein samples were digested by Filter Aided Sample Preparation (FASP) Method (*Wisniewski et al., 2009*). The tryptic peptides were separated by nanoflow liquid chromatography and analyzed by tandem mass spectrometry (Thermo Electron Finnigan). The LTQ-Orbitrap equipped with an NSI nanospray source (1.5 kV) was operated in data-dependent mode, in which the normalized collision energy was 35%. Full scan was detected in the Orbitrap analyzer (R=60,000 at m/z 300) followed by MS/MS acquisition of the ten most-intense ions in LTQ. Mass calibration used an internal lock mass (m/z 445.120025), the dynamic exclusion repeat count was 1, the repeat duration was 30 s, and the exclusion duration window was 120 s. Raw Orbitrap full-scan MS and ion trap MS2 spectra were processed by MaxQuant 1.3.0.5 (*Cox and Mann, 2008*). A composite target-decoy database was created with the program Sequence Reverser from the MaxQuant package. All identified MS/MS spectra were searched against this target/decoy database (Human UniProtKB/Swiss-Prot database, 2014-10-29 download). Spectra were initially searched with a mass tolerance of 7 ppm in MS and 0.5 Da in MS/MS and strict trypsin specificity. Cysteine carbamidomethylation was searched as a fixed modification, whereas N-acetyl protein, oxidized methionine, and mono-methylated lysine (putative methylation site) were searched as variable modification. The estimated false discovery rate (FDR) of all peptide and protein identifications was fixed at maximum 1% (*Olsen et al., 2006*).

## In vitro methylation assay

Prepare a 100 µl reaction mixture containing 75 µl MAB buffer (50 mM Tris pH8.5, 20 mM KCl, 10 mM MgCl$_2$, 10 mM BME, 250 mM Sucrose), 4 µl of $^3$H labelled S-adenosyl methionine ($^3$H-SAM) (Perkin elmer), 4 µl of GST-Set7 protein (at 1 mg/ml), 10 µl 10×BSA and 4 µl of 10×protease inhibitor cocktail. Add 20 µl Mix contents in five new tubes. Transfer 5 µl (5 mg/ml) purified MBP, MBP-K436, MBP-K436R, MBP-K595 and MBP-K595R into the tubes, respectively. Pipette up and down gently and incubate the tubes at 37°C for overnight. To stop the reaction, add 8 µl of 4×SDS-PAGE loading buffer to each tube and heat the samples at 95°C for 5 min. Then run the SDS-PAGE gel and dry the gel. Put the dried gel into the film cassette without the wrap and a piece of Kodak BioMax MS film (Sigma). Put the cassette at -80°C freezer for appropriate time and develop the film.

## RNA interference

Sequences of RNAi oligonucleotides are as follows:
Nonspecific small interfering RNA (siRNA): UUCUCCGAACGUGUCACGU
Set7 siRNA-1 sense strand: CCUUUGAUCUGUAUCUCUUTT
Set7 siRNA-2 sense strand: GGACCUAAUACUGUUAUGUTT
Gli3 siRNA-1 sense strand: CCCGUGGGUAUGUCUAUAUTT
Gli3 siRNA-2 sense strand: GCUCUAAGUAGGUAUUUAATT

 All RNAi oligonucleotides were purchased from Shanghai GenePharma Company, China. These RNAi oligonucleotides were transfected into cells by using the Lipofectamine RNAi mix transfection kit (Invitrogen) according to the manufacturer's instructions. The siRNA in paper represent siRNA-1.

 Set7 shRNA-1 sense strand: CCGGCCGTGTTCAGAGATACCAAATCTCGAGTATCTCTGAA-CACGGTTTTTG; Set7 shRNA-1 antisense strand: AATTCAAAAAACCGTGTTCAGAGATACCAAATC TCGAGTCTCT GAACACGG; Set7 shRNA-2 sense strand: CCGGCCTAATACTGTTATGTCGTTTC

TCGAGAAACGACATAACAGTATTAGGTTTTTG; Set7 shRNA-2 antisense strand: AATTCAAAAACC TAATACTGTTATGTCGTTTCTCGAGAAACGACATAACAGTATTAG.

This shRNA was cloned into the PLKO vector and transfected into HEK293T cells to make pseudoviruses. Then, the pseudoviruses were collected to infect the indicated cells.

## Tissue sample preparation

The lysis tubes (Precellys) were prepared and added 200 μl denature buffer (50 mM Tris-HCl, 0.5 mM EDTA, 1% SDS and proteinase inhibitor cocktail) before use. The mouse embryos at 14.5 dpc were dissected and carefully put the indicated tissues into these tubes until all the embryos were dissected. Using the Precellys Homogenizers to homogenate tissues. Add 4×SDS loading buffer before western blot.

## Luciferase assay

NIH-3T3 cells cultured in 24-well plates were co-transfected with 1.15 μg 8×GliBS luciferase reporter, 115 ng pRL-TK Renilla, 230 ng indicated constructs per well. 48 hr after transfection, the luciferase activity was measured by Dual-GloTM luciferase assay system (Promega) in triplicate. Statistical significance was determined using student t-test.

## Real-time PCR

Cells were lysed in TRIzole (Invitrogen) and then performed the RNA isolation following the standard protocol. An optimized amount of RNA was used for Reverse transcription using ReverTra Ace qPCR RT Master Mix with gDNA Remover (TOYOBO, Japan). Real-time PCR was performed on the CFX96 TouchTM Real-Time PCR Detection System (BIO-RAD) with SYBR Green Real time PCR Master Mix (TOYOBO, Japan). 2-Ct method was adopted to quantify the results. Statistical significance was determined using student t-test. The primer pairs used for real-time PCR were listed below.

Set7_Mus_RT_F: CACTCCTTCACTCCGAACTG
Set7_Mus_RT_R: TTCAGCTCCACTTGATACCAC
Gli3_Mus_RT_F: AGAAGCCCATGACATCTCAG
Gli3_Mus_RT_R: GGTCTGCTACACTACCTCCA
Gli1_Mus_RT_F: CTGAGACGCCATGTTCAATCC
Gli1_Mus_RT_R ACCAGAAAGTCCTTCTGTTCCC
Ptch1_Mus_RT_F: ACTACCCGAATATCCAGCACC
Ptch1_Mus_RT_R: ATCCTGAAGTCCTTGAAGCCA
GAPDH_Mus_RT_F: GAGAAACCTGCCAAGTATGATGAC
GAPDH_Mus_RT_R: TGGAAGAGTGGGAGTTGCTG
Set7_homo_RT_F: CATTTCTACCAATGCTCTTCTTCC
Set7_homo_RT_R: ACATAACAGTATTAGGTCCCACAG
Gli1_homo_RT_F: GGGATGATCCCACATCCTCAGTC
Gli1_homo_RT_R: CTGGAGCAGCCCCCCCCAGT
Ptch1_homo_RT_F: CCACAGAAGCGCTCCTACA
Ptch1_homo_RT_R: CTGTAATTTCGCCCCTTCC
GAPDH_homo_RT_F: GAGTCAACGGATTTGGTCGT
GAPDH_homo_RT_R: GACAAGCTTCCCGTTCTCAG

## Immunofluorescence

NIH-3T3 cells were plated in 24-well plates, and fixed in 4% formaldehyde for 20 min. Cells were permeabilized with PBS/0.5% Triton X-100 for 3 min and nonspecific binding sites were blocked with 2% BSA in PBST (PBS with 0.5% Tween-20). Cells were stained with primary antibodies diluted in 2% BSA/PBST for 1 hr at room temperature. After washing four times with PBST, cells were incubated for 1 hr with appropriate secondary antibodies together with 1 g/ml of 4,6-diamidino-2-phenylindole (DAPI) in 2% BSA/PBST. Leica LAS SP5 confocal microscope was employed to take images. Primary antibodies used in this study: Mouse anti-ac-tubulin (Sigma). Secondary antibodies used in this study were bought from Millipore Company.

## ChIP assay

Cells were cross-linked for 10 min at 37°C by adding formaldehyde to a final concentration of 1%. The cross-linking was stopped by adding glycine to a final concentration of 0.125 M. Fixed cells were then washed with PBS, and sonicated in sonication buffer (50 mM Hepes-KOH, pH7.5, 140 mM NaCl, 1 mM EDTA, pH8.0, 1% Triton X-100, 1% sodium deoxycholate, 0.1% SDS, and proteinase inhibitor cocktail) with a Bioruptor Sonicator to yield genomic DNA fragments with size of about 250bp. Lysates were centrifuged, collected and incubated with M2 beads overnight on a rotator at 4°C. Beads were washed 4 times with ChIP wash buffer (0.1% SDS, 1% Triton X-100, 2 mM EDTA, pH8.0, 150 mM NaCl, and 20 mM Tris-Cl, pH8.0) and finally washed with ChIP final wash buffer (0.1% SDS, 1% Triton X-100, 2 mM EDTA, pH8.0, 500mM NaCl, and 20 mM Tris-Cl, pH8.0). Genomic DNA was eluted with elution buffer (1% SDS and 100 mM NaHCO3) at 65°C for 30 min. 5 M NaCl was add to a final concentration of 200 mM for a further incubation at 65°C for 4 hr or overnight. Proteins were then removed by 0.25 mg/ml proteinase K in a 5 mM EDTA solution after 2 hr incubation at 55°C. Genomic DNA was purified using DNA purification kit (QIAGEN) for Real-time PCR. Statistical significance was determined using student t-test. Primer pairs used in this study are listed below.
a-Forward: GGAGAGCAATTAGGAAGTTTGG
a-Reverse: GGAGAGCAATTAGGAAGTTTGG
b-Forward: TCTCTAGCTTCTATCCACCCA
b-Reverse: TCTCTAGCTTCTATCCACCCA
c-Forward: GATTGGACTCCTGACCTGTG
c-Reverse: CATGTTAGGAAACCCACCCA
*Hhip*-Forward: CTATATTAAGCCCAGACTTTCCAG
*Hhip*-Reverse: CATCCCAGTGCAATGTTCAG
*Ptch1*-Forward: TGGAGGGCAGAAATTACTCAG
*Ptch1*-Reverse: TAATGGAAGTATTGCATGCGAG

## MTT assay

Cells were seeded in 96-well plates at 5000 cells/well. Relative cell growth rate was determined by MTT assay daily. 20 µl of MTT working solution (5 mg/ml) was added into each well and incubated for 4 hr. Then discard the supernatant and dissolve the MTT formazan in 100 µl DMSO. The absorbance was measured at the wavelengths of 595 nm and 630 nm. Statistical significance was determined using student t-test.

## Wound healing assay

Cells ($5 \times 10^5$) were seeded in 3.5 cm dishes and incubated overnight. The cell monolayers were scratched horizontally using a sterile 10 µl pipette tip. The floating cells were removed with PBS and cultured again in DMEM without serum. Photographic images along the scrape line were obtained at indicated time points by an inverted microscope.

## Soft agar assay

Briefly, 1ml culture medium with 1% agar was first plated into 3.5 cm dishes. After the bottom agar became solidified, each well received another 1 ml 0.3% agar in culture medium carrying 5000 cells. Cells were covered with 1 ml of complete medium and medium was changed every four days. Colonies were stained with 0.005% crystal violet and counted after 3-weeks of incubation.

## Transwell migration assay

The transwell migration assay was carried out using a 24-well transwell cell culture chamber (Corning, New York) with an 8 µm pore size. Briefly, $1 \times 10^4$ cells were added into an upper chamber with 200 µl DMEM without serum. 700 µl DMEM containing 20% FBS was added into the lower chamber as a chemo attractant. After 48 hr, the non-migration cells were manually removed with a rubber swab. Cells that migrated to the lower side of the membrane were stained with crystal violet and photographed using an inverted microscope.

## Transwell invasion assay

The transwell invasion assay was carried out using a 24-well transwell cell culture chamber (Corning, New York). The 8 μm pore size upper chambers were coated with growth factor-reduced Matrigel (BD). Briefly, A549 cells were incubate in serum free medium for 24 hr. Matrigel were diluted three times with serum free DMEM. Add 50 μl above mixture to the upper chamber and incubate for at least 4 hr. Then $2 \times 10^4$ cells were added into an upper chamber with 200 μl DMEM without serum. 700 μl DMEM containing 20% FBS was added into the lower chamber as a chemoattractant. After 48 hr, the non-invasion cells were manually removed with a rubber swab. Cells that invade to the lower side of the membrane were stained with crystal violet and photographed using an inverted microscope.

## Tumor formation assay

Female nude mice were obtained from Shanghai Experimental Animal Center and maintained in pathogen-free conditions. Exponential phase A549 cells were trypsinized and washed with PBS (HyClone) and suspended in fresh PBS in 4°C. $5 \times 10^6$ cells were injected subcutaneously into four-week-old female nude mice on both left and right flank. Tumor size was measured every 7 days, and tumor volume was estimated using the formula: tumor volume=$0.5 \times$length$\times$width$^2$. At the end of this experiment, tumors were harvested, weighed and photographed. All procedures for animal experimentats were performed in accordance with the Institutional Animal Care and Use Committee guidelines of the Animal Core Facility of the Institutes of Biochemistry and Cell Biology (SIBCB). The approval ID for using the animals was 087 by the Animal Core Facility of SIBCB.

All the experiments were repeated more than twice.

## Acknowledgements

We are grateful to Drs. Chi-chung Hui, Xueliang Zhu, Hongbin Ji and Min Wu for reagents. We thank Drs. Rong Zeng, Catherine CL Wong, Min Huang and Chen Li for support in mass spectrometry assay. This work was supported by grants from the National Natural Science Foundation of China (31171414, 31371492, 81372233, XDB19020104).

## Additional information

### Funding

| Funder | Grant reference number | Author |
|---|---|---|
| National Natural Science Foundation of China | 31171414 | Yun Zhao |
| National Natural Science Foundation of China | 31371492 | Yun Zhao |
| National Natural Science Foundation of China | 81372233 | Hailong Wu |
| The "Strategic Priority Research Program" of the Chinese Academy of Sciences | XDB19020104 | Yun Zhao |

The funders had no role in study design, data collection and interpretation, or the decision to submit the work for publication.

### Author contributions

LF, Conception and design, Acquisition of data, Analysis and interpretation of data, Drafting or revising the article; HW, DG, LZ, Conception and design, Analysis and interpretation of data, Drafting or revising the article; SYC, Conception and design, Drafting or revising the article, Contributed unpublished essential data or reagents; YZ, Conception and design, Analysis and interpretation of data, Drafting or revising the article, Contributed unpublished essential data or reagents

### Author ORCIDs

Yun Zhao, http://orcid.org/0000-0002-7807-6094

## Ethics

Animal experimentation: All procedures for animal experimentation were performed in accordance with the Institutional Animal Care and Use Committee guidelines of the Animal Core Facility of the Institutes of Biochemistry and Cell Biology (SIBCB). The approval ID for using the animals was 087 by the Animal Core Facility of SIBCB.

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
