## [Decision Letter]

Thank you for submitting your article "Set7 mediated Gli3 methylation plays a positive role in the activation of Sonic Hedgehog pathway" for consideration by *eLife*. Your article has been reviewed by one peer reviewer, and the evaluation has been overseen by Michael Green as the Reviewing Editor and Kevin Struhl as the Senior Editor.

The reviewers have discussed the reviews with one another and the Reviewing Editor has drafted this decision to help you prepare a revised submission.

Summary:

Fu et al. investigate the potential methylation regulation of Hh-Gli signaling. They show that Gli3 can be methylated at K436 and K595 by the lysine methyltransferase Set7, leading to enhanced stability and DNA binding activity. The results linking posttranslational methylation to Gli function are interesting and have potential cancer relevance. However, the authors need to address several issues to confirm their conclusions and demonstrate their physiological relevance.

Essential revisions:

1) All of the RNAi experiments need to be done with at least two unrelated siRNAs or shRNAs. This is done in some but not all experiments. For example, the use of two siRNAs or shRNAs appears to be missing in Figures: 1G, 2G, K and 3A, B, H.

2) The authors claim that Set7 methylates full-length Gli3, but not the truncated repressor form. However, this is hard to assess in several of the current immunoblots. High-quality immunoblots of Figure 1 should be provided. In addition, the authors should also test whether Set7 can methylate the Gli3 truncated form when ectopically expressed.

3) With the available K436/K595 antibodies, the authors should examine whether methylation at these sites occurs in vivo in responsive tissues in mice or fly. Please let us know if you have access to these tissue samples and the time frame for you to complete an experimental test of Gli3 methylation in vivo.

4) Figure 3. ChIP-qPCR should be carried out on other well-established Gli targets in addition to the sites in the Gli1 promoter.

Desirable but not essential revisions:

1) Is the level of methylation regulated by Hedgehog ligand input?

2) The importance of the manuscript would be strengthened if the authors could demonstrate that the Set7-Gli3 regulation occurs in canonical Hh tumors, such as BCC and MB.

3) Gli3 has been reported to function both as a transcriptional activator and a repressor. Therefore, the strong pro-oncogenic effect of Gli3 in the presented xenografts experiments may be a bit surprising. Can the authors comment on this issue?

---

## [Author Response]

Essential revisions:

1) All of the RNAi experiments need to be done with at least two unrelated siRNAs or shRNAs. This is done in some but not all experiments. For example, the use of two siRNAs or shRNAs appears to be missing in Figures: 1G, 2G, K and 3A, B, H.

We thank the Reviewers for this professional and constructive suggestion. As suggested, we have now included an additional siRNA or shRNA in revised Figures: 1G, 2G, K and 3A, B, H. In addition, following this suggestion, we made a double check of our manuscript and found that we also inappropriately used a single siRNA or shRNA for experiments of Figure 3, and original Figure 3—figure supplement 7 and 8. We have now corrected them by including two unrelated siRNA or shRNA in revised Figure 3, Figure 3—figure supplement 9 and 10.

2) The authors claim that Set7 methylates full-length Gli3, but not the truncated repressor form. However, this is hard to assess in several of the current immunoblots. High-quality immunoblots of Figure 1 should be provided. In addition, the authors should also test whether Set7 can methylate the Gli3 truncated form when ectopically expressed.

We are sorry for the low-quality immunoblots in our original manuscript. As suggested, we have now repeated the experiments and replaced the Figure 1 with high-quality immunoblots. Thanks for point out this issue.

We also tested whether Set7 can methylate Gli3 truncated form in the context of ectopic expression. As shown in Figure 5, the methylation signals in Gli3 truncated form were detectable when Gli3^R^ or both Set7 and Gli3^R^ were ectopically expressed. As we showed in our manuscript, however, the methylation signals were undetectable in the Gli3 truncated form derived from either endogenous Gli3 or flag-Gli3 full length (please see the original Figure 1 and revised Figure 1). More important thing is that this phenomenon is repeatable. Thus we believe that the methylation signals in Figure 5 are quite artificial which are possibly due to excessive expression of Gli3^R^ or the lack of Gli3 processing in this ectopic expression model. Therefore, we decide not to include this result into the revised manuscript.

Author response image 1.Methylation signals in Gli3^R^ when both Set7 and Gli3^R^ were overexpressed.**DOI:**
http://dx.doi.org/10.7554/eLife.15690.020

3) With the available K436/K595 antibodies, the authors should examine whether methylation at these sites occurs in vivo in responsive tissues in mice or fly. Please let us know if you have access to these tissue samples and the time frame for you to complete an experimental test of Gli3 methylation in vivo.

We thank Reviewers for this good suggestion. Given that Shh signaling plays an important role in the development of multiple embryonic tissues of mice (Ingham & McMahon, 2001), we started to examine whether these methylation signals exist in the Gli3 in several responsive mouse embryonic tissues, such as brain, lung, skin, skeleton and gut, by western blot. These embryonic tissues were collected from embryos at 14.5 (dpc). Gli3 methylation signals were detected by western blot using Me-K436 and Me-K595 antibodies respectively. The methylation signals on Gli3 full length were only detected in embryonic lung tissues. Consistent with our original manuscript, methylation signals were not detectable in Gli3 truncated forms of all embryonic tissues tested. Accordingly, we also included this data into revised Figure 1—figure supplement 4.

4) Figure 3. ChIP-qPCR should be carried out on other well-established Gli targets in addition to the sites in the Gli1 promoter.

We thank the Reviewers for this constructive suggestion to improve our study. *Hhip* and *Ptch1* are well-established Shh downstream target genes. A previous report indicates that Gli3 can bind *Ptch1* promoter to transactivate *Ptch1 (*Agren et al., 2004). Bioinformatic analysis revealed a putative binding sequence of Gli3 in the promoter of *Hhip*. Through CHIP-qPCR, we found that Gli3 can bind to promoter regions of both *Hhip* and *Ptch1* and this binding is subjected to the regulation of Set7. Accordingly, we also included this data into revised Figure 3—figure supplement 3.

Desirable but not essential revisions:

1) Is the level of methylation regulated by Hedgehog ligand input?

We do observe increased methylation levels in endogenous Gli3 after Shh treatment in 3T3 cells. This Shh mediated the increase of methylation signals may be probably due to increased Gli3 full-length or/and Set7 levels which were presented in Figure 2. Therefore, we conclude that the Gli3 methylation level is regulated by Hedgehog ligand input. Accordingly, we also included this data in revised Figure 2—figure supplement 2. Thanks!

2) The importance of the manuscript would be strengthened if the authors could demonstrate that the Set7-Gli3 regulation occurs in canonical Hh tumors, such as BCC and MB.

This is a very good suggestion. In our opinions, it’s very likely that the Set7-Gli3 regulation occurs in canonical Shh tumors. Unfortunately, due to the lack of collaboration with clinicians, we do not have BCC and MB samples to examine the existence of this regulation in these tumors currently. However, we will definitely pursue this point in our future studies. Thanks for this important suggestion. Thanks!

3) Gli3 has been reported to function both as a transcriptional activator and a repressor. Therefore, the strong pro-oncogenic effect of Gli3 in the presented xenografts experiments may be a bit surprising. Can the authors comment on this issue?

We agree with the reviewers’ comment that Gli3 functions both as a transcriptional activator and a repressor. Because of this bipartite feature of Gli3, its oncogenic potential has been long neglected. However, recent emerging evidence indicates that Gli3 actually promotes tumor progression in multiple cancer types such as cervical cancer and colon cancer (Kang et al., 2012; Wen et al., 2015). Similar to these two cancer types, we demonstrate the oncogenic role of Gli3 in lung cancer in our current study. The tumor promoting role of Gli3 may be probably attributed to its transactivation of *Gli1* or other oncogenes.